# Recurrent Translocations in Topoisomerase Inhibitor-Related Leukemia Are Determined by the Features of DNA Breaks Rather Than by the Proximity of the Translocating Genes

**DOI:** 10.3390/ijms23179824

**Published:** 2022-08-29

**Authors:** Nikolai A. Lomov, Vladimir S. Viushkov, Sergey V. Ulianov, Alexey A. Gavrilov, Daniil A. Alexeyevsky, Artem V. Artemov, Sergey V. Razin, Mikhail A. Rubtsov

**Affiliations:** 1Department of Molecular Biology, Faculty of Biology, Lomonosov Moscow State University, Moscow 119234, Russia; 2Institute of Gene Biology, Russian Academy of Sciences, Moscow 119334, Russia; 3A.N. Belozersky Institute of Physico-Chemical Biology, Lomonosov Moscow State University, Moscow 119234, Russia; 4Department of Neuroimmunology, Center for Brain Research, Medical University of Vienna, 1090 Vienna, Austria; 5Department of Biochemistry, Center for Industrial Technologies and Entrepreneurship I.M., Sechenov First Moscow State Medical University (Sechenov University), Moscow 119435, Russia

**Keywords:** therapy-related AML, t-AML, chromosomal translocation, topoisomerase inhibitors, etoposide, topoisomerase inhibitor-related leukemia, *AML1*, *RUNX1*, *MLL*, *KMT2A*

## Abstract

Topoisomerase inhibitors are widely used in cancer chemotherapy. However, one of the potential long-term adverse effects of such therapy is acute leukemia. A key feature of such therapy-induced acute myeloid leukemia (t-AML) is recurrent chromosomal translocations involving *AML1 (RUNX1)* or *MLL (KMT2A)* genes. The formation of chromosomal translocation depends on the spatial proximity of translocation partners and the mobility of the DNA ends. It is unclear which of these two factors might be decisive for recurrent t-AML translocations. Here, we used fluorescence in situ hybridization (FISH) and chromosome conformation capture followed by sequencing (4C-seq) to investigate double-strand DNA break formation and the mobility of broken ends upon etoposide treatment, as well as contacts between translocation partner genes. We detected the separation of the parts of the broken *AML1* gene, as well as the increased mobility of these separated parts. 4C-seq analysis showed no evident contacts of *AML1* and *MLL* with loci, implicated in recurrent t-AML translocations, either before or after etoposide treatment. We suggest that separation of the break ends and their increased non-targeted mobility—but not spatial predisposition of the rearrangement partners—plays a major role in the formation of these translocations.

## 1. Introduction

Targeting topoisomerase II is an effective treatment approach in cancer chemotherapy. DNA topoisomerase II (TopoII) plays a critical role in replication and transcription, where it resolves DNA catenanes and relieves the torsional stress in DNA. During the catalytic cycle, TopoII generates double-strand breaks (DSBs) in one DNA duplex and transits the second one through the DSB. Following the transit, the DSB is re-ligated [1]. The poisoning of TopoII leads to inhibition of both the re-ligation function of the enzyme and the accumulation of multiple DSBs in dividing cancer cells [2,3,4,5]. However, a side effect of this therapy is the formation of DSBs in other proliferating cells such as those in bone marrow. The ends of two DSBs could be ligated by the non-homologous end-joining (NHEJ) mechanism, which leads to the formation of chromosomal translocations [6] associated with an increased risk of leukemia [7]. These topoisomerase inhibitor-related (TI-related) leukemias occur as a late complication of therapy after a 1–2 year latency period [8,9] and often present as overt acute myeloid leukemia (AML) with poorer outcomes compared to de novo AML [10,11,12]. A characteristic feature of tumor cells in TI-related AML is the presence of balanced chromosomal translocations, most often involving the *KMT2A* gene (also known as *MLL*) at 11q23.3 and the *RUNX1* gene (also known as *AML1)* at 21q22.1 [13,14]. The *KMT2A* (histone-lysine N-methyltransferase 2A) gene encodes a protein that methylates histone H3 on lysine 4 (H3K4) to promote genome accessibility and transcription [15,16]. The *RUNX1* (runt-related transcription factor 1) gene encodes a transcription factor that regulates hematopoiesis [17]. Recurrent translocations involving *AML1* and *MLL* genes lead to the formation of fusion proteins that exert their leukemogenic effects through the dysregulation of genes involved in cell proliferation [18].

Both *AML1* and *MLL* genes have a limited set of recurrent translocation partners in TI-related AML (Table 1).

It is worth noting that this set of translocations differs from the recurrent translocations in other types of t-AML, such as those induced by alkylating agents or radiotherapy [10,19]. Therefore, the manner in which the DSBs are introduced plays a significant role in the formation of translocations. However, the mechanisms responsible for the formation of recurrent translocations in TI-related AML are poorly understood.

Spatial juxtaposition of translocating loci before or after the formation of DSBs is a prerequisite for effective translocation. However, fluorescence in situ hybridization (FISH) and chromatin conformation capture-based techniques reveal that interphase chromosomes that occupy separate regions of the nucleus and loci on the same chromosome tend to be in spatial proximity to each other [33,34,35]. The formation of chromosomal rearrangements is influenced by this territorial organization of interphase chromosomes [36].

Data from high-throughput chromosome conformation capture (Hi-C) revealed a correlation between the spatial proximity of loci and the probability of translocation between them [37]. Many translocation-prone pairs of regions genome-wide, including the translocation partners *BCR-ABL* and *MYC-IGH*, display elevated Hi-C contact frequencies in normal human cells [38]. FISH experiments also showed that close association of neighboring chromosomes in lymphocytes favors myeloma-associated translocations [39]. However, these observations do not exclude the formation of the translocations between initially distant loci as the result of increased mobility after DSB formation. It has been shown that locus mobility increases following DSB formation and DSBs cluster together [40]. Moreover, in vivo imaging of individual loci demonstrates the formation of translocations between initially distant loci [41,42].

In the present study, we investigated the mobility and spatial proximity of translocating loci involved in rearrangements associated with TI-related AML. As previously shown, treatment of lymphoblastoid Jurkat cells with the Topo II inhibitor etoposide leads to separation of the ends of DSB in the *MLL* gene, which is frequently followed by relocation of the broken ends beyond the chromosomal territory [43]. In our study, similar observations were made for the *AML1* gene. Using FISH on three-dimensionally preserved nuclei (3D-FISH), we showed that exposure of Jurkat cells with etoposide caused separation of the ends of DSBs in the *AML1* gene, and frequent relocation of broken ends beyond the chromosomal territory. To search the genome for sequences contacting *MLL* or *AML1*, we used a 4C-seq method (chromosome conformation capture followed by sequencing). However, we did not detect spatial proximity of *MLL* or *AML1* to their translocation partners either before or after etoposide treatment. This finding suggests that spatial proximity does not play a substantial role in determining patterns of *AML1* and *MLL* translocation partners. Occasional encounters of various translocation partners in cells exposed to etoposide are likely to occur due to the increased mobility of broken DNA ends, whereas the specific patterns of translocations in AML could originate as the result of the selection of cellular clones [44].

## 2. Results

Our task was to analyze whether etoposide induces DSBs in the *AML1* gene and whether the ends of the DSBs move outside the territory of chromosome 21 where they are normally located. We used the FISH technique which enables the detection of DSBs over a wide interval within the gene. Considering the infrequency of the events we were studying, we combined FISH with automated analysis of the results to obtain relevant statistics. Jurkat cells (cultured human lymphoid cells) were used as a convenient cell culture for 3D-FISH.

The subsequent task was to determine whether a spatial predisposition of *MLL* and *AML1* genes to the translocation in non-treated cells occurs and to discover if there is a directed motion of the *MLL* and *AML1* genes to certain loci after treating the cells with etoposide. To analyze the *MLL* and *AML1* genome-wide chromatin contacts, we used the 4C-seq method.

### 2.1. Etoposide Treatment Leads to the Formation of DSBs in AML1 Followed by Separation of the Break Ends

To verify that etoposide causes DSBs in the *AML1* gene and to determine where the broken ends are located relative to the territory of chromosome 21, we treated Jurkat cells with etoposide and performed 3D-FISH with a far-red whole chromosome 21 painting probe and a dual color break-apart (DC BA) *AML1* probe, the colors of which correspond to upstream and downstream regions of the *AML1* gene. For the quantitative detection of rare events like *AML1* breaks, we used custom software, called Conredet, which reconstructs 3D images of the nuclei from Z-stacks of confocal 2D images (see the Materials and Methods section). The program recognizes fluorescent signals of each color and determines the coordinates of the regions of the *AML1* gene, as well as the boundaries of the chromosomal territories and cells (Figure 1). Conredet also identifies which regions of *AML1* belong to the same allele and applies quality filters: only cells with 2 alleles of the *AML1* gene and 2 chromosomal territories (which could merge into one) were used for the downstream analysis (see the description of the algorithm in the Materials and Methods section). The data obtained are in Appendix A.

First, we plotted the distribution of the pairwise distances between two regions of the *AML1* gene in the untreated (control) cells (Figure 2A). The 99th quantile of the distribution was designated as “broken” alleles, since we demonstrated that beyond this value there is a much larger percentage of alleles in the cells treated with etoposide than in the control cells (Figure 2B).

We subsequently analyzed the cells treated with etoposide (2 replicates of each series in the experiment). In these cells, the percentage of split signals was higher, which indicates the “broken” alleles.

The data confirm that, as in the case of the *MLL* gene [43], etoposide induces DSBs in the *AML1* gene, and the ends of some DSBs move apart (i.e., in 3% of total *AML1* alleles under current experimental conditions).

To find out whether the induction of DSBs in *AML1* by etoposide differs from the induction of DSBs by ionizing radiation (IR), we performed an experiment with X-rays. Opposite to etoposide treatment, exposure of cells to even a high irradiation dose (7 Gy) did not increase the number of “broken” *AML1* compared to control cells (Figure 2C). Thus, separation of the *AML1* DSB ends is a characteristic of the etoposide-induced DSBs.

### 2.2. The Distribution of AML1 Alleles Relative to the Chromosomal Territory Boundaries Changes after Etoposide Treatment

Using the Conredet program, we studied the distances between regions of one allele and the distance to the chromosomal territory boundary for each region of *AML1*. The results are reflected in the scatterplots (Figure 3).

We found that the percentage of broken alleles localized outside the chromosomal territory was significantly higher (~2.9 times, *p* < 0.05 in the chi-square test) than that for non-broken alleles. However, we could not guarantee that all “non-broken” alleles in etoposide-treated cells were actually intact. Therefore, we compared broken alleles from etoposide-treated cells with intact alleles from non-treated cells, and the difference was more evident (~5.8 times, *p* < 0.001 in the chi-square test; see statistic in Appendix A).

### 2.3. MLL and AML1 Do Not Demonstrate Increased Contact Frequency with Their TI-Related Rearrangement Partners in Both Control and Etoposide-Treated Cells

To determine whether there is a spatial predisposition of genes rearranged in TI-related AML, we performed a 4C-seq analysis. DNA-protein complexes were crosslinked with formaldehyde, and DNA was digested using HindIII, diluted and ligated. The resulting DNA fragments, which are fusions of physically associated in 3D space DNA regions, were self-circularized by DpnII-treatment and ligation. Then, circular DNA molecules were amplified by reverse PCR with primers designed to loci of interest (4C bait loci). This 4C library was sequenced with Illumina to find the sequences contacting with bait loci, that have been selected on the *AML1, MLL,* and *CCND1* genes (Appendix A). The *CCND1* gene was chosen as a control because it is not involved in chromosomal translocations and is localized on the same chromosome 11 as the *MLL* gene. We also performed the same 4C-seq analysis of cells treated with etoposide to determine if a spatial proximity to translocations occurs as a result of etoposide treatment. 4C libraries for all biological conditions were prepared in two independent replicates. We obtained 7–20 million uniquely mapped reads per library, and because the replicates demonstrated a high correlation (Pearson’s r > 0.999, *p*-value < 2.2 × 10^−16^, Appendix A), we combined them after normalization to the total library size (Appendix A).

4C-seq profiles were visualized using the UCSC genome browser. First, we do not observe the pattern that *AML1* or *MLL* contact more often with their partner genes. This observation denotes the absence of spatial predisposition for the rearrangements associated with TI-related AML. Second, we also did not observe an increase in the number of *AML1-* and *MLL*-centered contacts with partner genes after etoposide treatment (Figure 4 and Figure 5). We therefore concluded that translocation partners do not spatially co-localize preferably with each other after induction of DSBs. To confirm the conclusions based on the 4C-seq profiles, we compared the number of *AML1* contacts (4C-seq signal) with its translocation partner genes and the average number of *AML1* contacts with each human gene. Translocation partners of *AML1* did not show significantly different levels of 4C-seq signal in *AML1* 4C libraries as compared to other human genes on non-bait chromosomes (*p*-value ranging from 0.35 to 0.97 in individual samples, Appendix A). Translocation partners of *MLL* also did not show altered 4C-seq signal in *MLL* 4C libraries as compared to other human genes on non-bait chromosomes (*p*-values ranging from 0.15 to 0.78, Appendix A). Finally, the 4C-seq data showed that most of the 4C-captured contacts are established in *cis*, which is consistent with previously published Hi-C data [34]. The proportion of the *trans*-contacts among all contacts did not change in etoposide-treated cells for *AML1,* and this proportion decreased by only 2.4% for *MLL* and 3.2% for *CCND1*. The latter observation is consistent with the FISH data, demonstrating that chromosomal territories do not expand and most *AML1* alleles do not leave their chromosomal territories following treatment with etoposide.

## 3. Discussion

Previous studies of chromosomal translocations have provided a variety of data that are sometimes contradictory. Some studies have demonstrated the importance of the spatial proximity of loci for translocation [37,38,39,41]. Other studies have shown the increased mobility of DSBs compared to that of intact loci [40,42,45]. Collectively, the results reported to date have been interpreted in the context of two hypotheses that emphasize either the importance of the initial position of the rearranged loci or the mobility of broken alleles. The “contact-first” hypothesis proposes that the misrepair of chromosome breaks can take place only when the breaks occur in colocalized chromatin fibers. The “breakage-first” hypothesis assumes that DSBs move across large distances in the nucleus before interacting, suggesting that the spatial predisposition of the loci is not necessary for rearrangement [46]. However, data obtained to date do not reveal a single mechanism that determines which translocation will occur. Rather, all factors affect the probability of translocation, in different situations and with varying degrees of probability [47]. For example, experiments on cells with a genomic loci visualization system in vivo showed that both close and distant loci form translocations after DSB induction, with this effect observed more frequently in the close loci [48]. In addition to the proximity of genes and the mobility of DSBs, the formation of certain translocations is determined by the probability of a break occurring at a particular site [37,49,50,51], as well as the origin of the break. For example, DSBs caused by topoisomerase poisons are more mobile than those caused by radiation [52], and sites at the anchors of chromatin loops are preferred for etoposide-induced DNA breaks [53,54]. It is apparent that some of these factors are related to each other; for example, the manner in which DSB is induced affects the probability of the break occurring at a given locus, its repair pathway, persistence time, and the number of DNA breaks at other loci.

As mentioned above, topoisomerase inhibitor-related leukemias are characterized by a limited set of recurrent translocations (Table 1). The selective advantage of translocations for cells plays a substantial role in the observation that we see this particular translocation in patients. However, the selective advantage is not the only factor, as different recurrent translocations are observed in other types of therapy-related leukemia, such as that induced by alkylating agents or radiotherapy [10,19]. In our work, we investigated the factors affecting the formation of recurrent translocations in TI-related AML, with a particular focus on the factors preceding the translocation, such as the frequency of translocation partner contacts and the dynamics of DSBs prior to the rearrangement.

It has been shown that *MLL* and *AML1* genes are susceptible to DSB damage caused by treatment with topoisomerase inhibitors [13,55,56,57,58]. In the present study, we have shown that exposure of cells to topoisomerase II inhibitor etoposide causes the *AML1* ends to move apart. This finding demonstrates the free movement of unrepaired DNA ends, although the ends of DSB do not typically move separately since their ends are held by the repair proteins that perform DNA end-bridging [59]. When single DSBs were induced by meganuclease, the ends of the DSBs did not usually separate [41,48]. Possibly, the unrepaired ends move apart if the cell repair systems are unable to cope with a large number of DSBs. Etoposide treatment results in many DSBs, which has been confirmed by comet assay [60] or by measuring γH2AX levels [61,62]. Another possible explanation for the split DSB ends is that topoisomerase II poisoning results in a permanent DSB, which persists for a lengthy amount of time because they require a special repair pathway: proteasomal degradation of the stalled enzyme and end-processing of DNA [3,63,64]. As a result, the ends of DSBs are not held together by proteins of the repair systems, which increases the likelihood of misrepair and translocation.

In case of IR-treatment we did not see *AML1* ends moving apart. We can suggest two explanations. First, unlike etoposide, IR should lead to DSBs that are more evenly distributed throughout the genome. When cells were treated even with a high dose of IR, we did not see DSBs specifically in *AML1*. Second, it is possible that IR-induced breaks are repaired faster than etoposide-induced breaks, which may explain the absence of separated parts of *AML1* in IR-treated cells.

The broken ends of *AML1* are more often localized outside the chromosome territory compared to intact alleles. This can be interpreted in terms of increased mobility of the broken ends in the nucleus space. This mobility may allow for occasional colocalization of genes that were originally located far apart in the nucleus.

The 4C-seq data suggest that there are no preferential contacts of *AML1* and *MLL* with their recurrent translocation partner genes. Therefore, for translocations associated with TI-related AML, spatial proximity is not a determining factor. This observation is consistent with data showing that chromosome conformation capture (3C) did not detect interactions between *AF9* or *AF4* with *MLL* [27]. Furthermore, the number of contacts between translocation partner genes does not increase relative to the number of contacts with other genes on non-bait chromosomes after etoposide treatment. Thus, we have not revealed directed movement of the broken translocation partner genes toward each other. Such directed movement would take place if the ends of DSBs moved to some nuclear compartment for repair, such as the nucleolus, for example, as previously suggested [65].

The results of both FISH experiments and 4C-seq show that, in the population of etoposide-treated cells, most of the *AML1* and *MLL* alleles still reside within their chromosomal territories. The resulting translocations can be explained by the long persistence time of these “wandering” ends of DSBs that require a substrate for repair, even if the repair is erroneous.

To summarize, the limited set of recurrent translocations in TI-related AML cannot be explained by spatial predisposition of translocation partner genes, but rather by the characteristics of DSBs and the functional role of the translocations. Such translocations give a clonal advantage to these cells, compared to others with different translocations, which surely occur, but are not found in cancer cells [44].

## 4. Materials and Methods

### 4.1. Cell Culture

The Jurkat human lymphoid cell line [66] was obtained from the collection of the Medical Genetics Research Center, RAMS. Cells were grown at 37 °C in a 5% CO_2_ incubator. RPMI 1640 medium supplemented with 10% FBS was used. Treatment of cells with etoposide was performed in the same medium at a concentration of 100 µg/mL for 1 h. Radgil (Gilardoni, Italy), a special medical X-ray unit for the irradiation of blood components, was used to irradiate the cells. The cells were irradiated for 6.5 min (7 Gy). The ionizing radiation doses for Jurkat cells have been selected based on literature [37,67].

### 4.2. 3D-FISH

3D-FISH was performed as previously described [43,68]. Briefly, cells were attached to glass coverslips coated with Cell-Tak™ (BD Bioscience) and fixed in 4% paraformaldehyde. Then, cells were permeabilized in 0.5% (*w*/*v*) Triton X-100/1x PBS, incubated for 12 h in 20% (*v*/*v*) glycerol/1x PBS, frozen four times in liquid nitrogen, treated with RNAse A (200 µg/mL) and equilibrated in 50% (*v*/*v*) deionized formamide for at least one week at +4 °C.

The *AML1* gene was visualized using fluorescently labeled bacterial artificial chromosome (BAC) probes (BlueGnome, UK). BAC RP11-299D9 is 140 kb long and labelled with AlexaFluor 488 green fluorophore, while BAC RP11-177L11 is 160 kb long and labelled with Cy3 orange fluorophore. Whole Chromosome 21 Painting probe (Human IDetect Chr 21 Paint Probe FAR RED) was labeled with far-red fluorophore IDYE 647. Denatured nuclei were hybridized with denatured probes for 48–72 h at 37 °C.

Images were acquired using a confocal laser scanning microscope LSM-510 Meta (Zeiss). Each experiment was captured in 10–15 microscopic fields. Each field was Z-stacked with a resolution of 1024 × 1024 and contained about 100 cells. A typical confocal series (Z-stack) contained 15–30 images with 400–800 nm intervals.

### 4.3. Image Processing

All Z-stacks obtained from a confocal microscope were exported as an image series and processed using our specifically tailored software called Conredet. The goal of image processing is to detect *AML1* alleles, 21 chromosomal territory, and whole nuclei as 3-dimensional objects and to measure the distances between them. The input to image processing is a Z-stack with three color channels, one each for the dual-color fluorescent probe, visualizing the two regions of *AML1* allele, and for the chromosomal territory. Nuclei were detected using background DNA fluorescence in one of the channels.

Image processing consists of the following steps: image pre-processing, nuclei detection, and detection of territory and DNA loci within each nucleus. Image pre-processing includes one filter: despeckling (3D 1 × 1 × 3 median filter). Nuclei are detected in an iterative process. First, the 3D gaussian blur (sigma = 1) is applied. Subsequently, the brightest voxel is selected as the nuclei center, and the cylindrical model of the nucleus is fit to the image to maximize total brightness within the cylinder using gradient descent and starting with the selected center. The nucleus is then removed from the image and the process is repeated starting from the next brightest voxel. To detect the DNA locus, a fixed number of the top brightest voxels is selected. Each locus is detected as a connected component on the graph of neighboring voxels. Chromosomal territory is detected using an identical process after the corresponding color channel is passed through a 3-dimensional Gaussian filter (sigma = 1).

To reduce object detection errors, several filters are applied to the resulting objects. First, for each kind of object, a threshold is defined for a minimal and maximal number of voxels in each object, and the objects not matching the threshold are removed. Following this procedure, only nuclei in which there are exactly two objects for each DNA locus and at least one detected territory are retained. The nuclei that do not match the nearest-locus criterion are removed. Signals with different colors between those centers which had the smallest possible distance in the cell were determined as paired signals. The nearest-locus criterion checks that the signals are correctly paired: 2 signals of different colors that are closest to each other are considered a pair. The criterion is defined as follows: if the green signal G1 is nearest to the red signal R1, then R1 is nearest to G1. The nearest-locus criterion removes the cells in which the loci were not detected. Instead, noise was detected as the DNA locus signal in the other location of the nucleus. All nuclei in which the distance between locus and pair (locus) was greater than 2000 nm were identified by the researchers in the images and considered incorrect detection results. All such cases were removed from the resulting data.

The resulting dataset was used to analyze distances between paired DNA loci and distances between DNA locus and the nearest chromosomal territory.

For the purposes of distance calculations, the DNA locus was treated as a single point that is the center of mass of the voxels detected as part of the locus, with voxel brightness defined as its weight. To calculate the distance between DNA locus and chromosomal territory, the distance between a DNA locus point and a plane containing three nearest points of chromosomal territory was determined using distance metrics as follows: distance is positive if the DNA locus is outside the chromosomal territory and negative if it is inside. All cells with broken alleles were identified in the images and the split signals were visually confirmed.

Cells with signals separated by 2000 nm or more from each other, or from the chromosomal territory, were not included in the analysis, as such examples are ultimately found to be recognition errors upon closer examination.

### 4.4. 3C and 4C-seq Libraries

The 4C-seq procedure was carried out as described in [69,70,71]. 4C-seq workflow and localization of the 4C-seq primers are in Appendix A. HindIII and DpnII were used as primary and secondary restriction enzymes, respectively. The PCR reactions for 4C library preparation were performed using the Expand Long Template PCR system (Roche) following the manufacturers’ protocol. The HindIII/DpnII 4C primers are as follows (5′-3′):
4C-AML1-HCAGGTGAGTGTGGAGGTAGAGAG4C-AML1-DCTTGGTTCCCCAGCTGAGAT4C-MLL-HATAGGCTCCATGTTGGCTCA4C-MLL-DCACAGGATACAAAGCAGAACTACTC4C-CCND1-HCCTGCCAACTTCGGTGTCC4C-CCND1-DAAGTTACCCGAGGCGGAGTC

(See Appendix A for the mapping 4C primers regarding the HindIII/DpnII sites on the *AML1* gene as example).

The PCR products were purified and concentrated using the QIAquick PCR purification kit (Qiagen). 4C libraries obtained with different pairs of primers were mixed in equal weights. Paired-end sequencing was performed on Illumina HiSeq 4000.

### 4.5. 4C-seq Analysis

4C-seq analysis was performed as previously described [70]. Briefly, sequencing reads originated from the bait sequence were trimmed and only the read containing the HindIII ligation site was chosen, whereas its pair (*read)* was discarded. The bait-originating subsequence of the read located before the HindIII site was used to classify the reads by the 4C anchor. Sequences located after the HindIII site were mapped to the reference genome hg19 using the bowtie aligner.

For every fragment between HindIII restriction sites, the number of reads that mapped to its left or right end was added and the resulting value was considered to be the raw 4C-seq signal for that restriction fragment. To explore contact frequencies of the genes being studied, the genome was partitioned into 500-kilobase non-overlapping windows and the number of fragments containing at least one 4C-captured contact was calculated for each window. Sequencing statistics for the 4C libraries are shown in Appendix A.

### 4.6. Statistics

To test the hypothesis of non-randomness of the tail on the histogram of distances between regions of one allele distribution after etoposide treatment, we used the following approach after formulating the following hypothesis: the true probability measure P of the tail is greater than zero. Based on the desired confidence level of 0.997, we accept the three-sigma rule; then, according to the theorem on the asymptotic of the empirical distribution [72], the true P differs from the experimental P* by no more than the value x = three sigmas divided by the square root of the total number of points, where sigma is the square root of the value P × (1 − P). If P * − x > 0, then with a reliability >0.997, the tail of the distribution has a strictly positive probability measure. If P is assumed to be approximately equal to the value P * = 0.029 calculated through the data, then for our case the tail detection is a relevant factor with reliability >0.997.

To test the hypothesis that the distribution of broken alleles localized outside/inside the chromosomal territory differs from that proportion for the non-broken alleles, we used the chi-square test (Appendix A).

A correlation matrix and plot were obtained for all 4C experiments (Appendix A). Pearson’s correlation coefficient showed a high correlation for the 4C replicates. The Mann–Whitney test was used to compare 4C-seq signal on translocation partners with 4C-seq signal on other human genes in all 4C libraries.

## 5. Conclusions

The following factors are responsible for recurrent translocations in TI-related AML. First, etoposide induces DSBs in the *AML1* and *MLL* genes. Second, some etoposide-induced DSBs persist for a lengthy period of time, and their ends split apart. Third, the split DSB ends are more mobile than the intact alleles, although determined movement has not been observed. Finally, there is no spatial proximity in translocating genes, so the determining factor of recurrent translocations is their selective advantage. In addition, since the mobility of DSBs induced by topoisomerase inhibitors has a role in chromosome translocation, the ability to manipulate the movements of the break ends could reduce the serious side effects of anticancer TI-therapy.

## Figures and Tables

**Figure 1 ijms-23-09824-f001:**
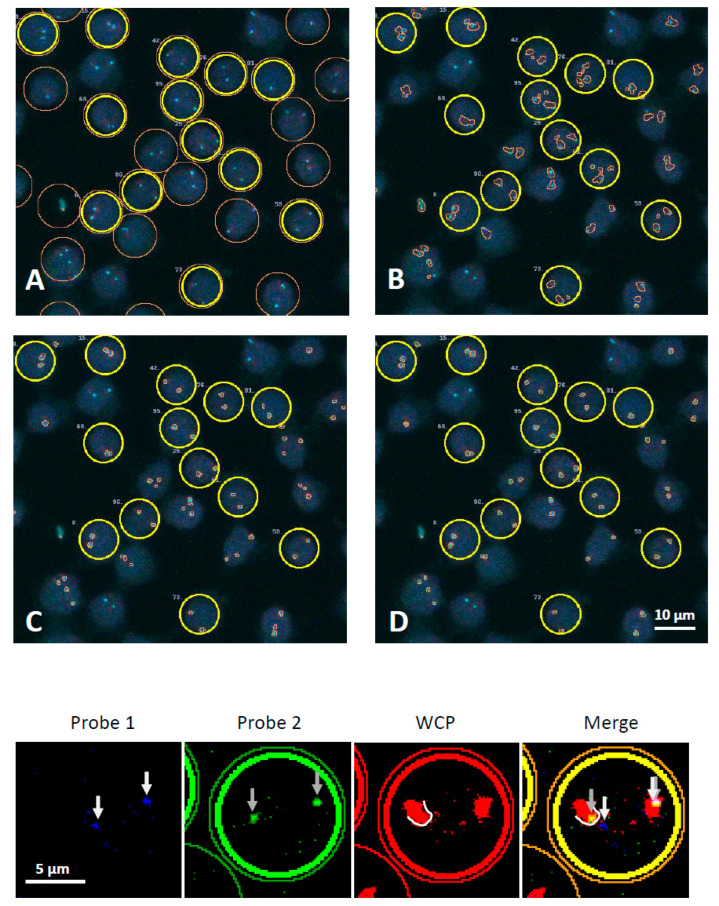
Example of program recognition of nuclei borders (**A**), chromosome territories (**B**), and upstream and downstream regions of the *AML1* gene (**C**,**D**). Merged images from 3 channels are shown. Bold circles indicate the cells used for analysis (which passed all quality filters). The lower panels show an example of a cell with a “broken” *AML1* allele (marked with arrows), one region of which is located outside the chromosomal territory visualized by a far-red whole chromosome painting probe (WCP). Probe 1 and probe 2 visualize upstream and downstream regions of the *AML1* gene (Appendix A).

**Figure 2 ijms-23-09824-f002:**
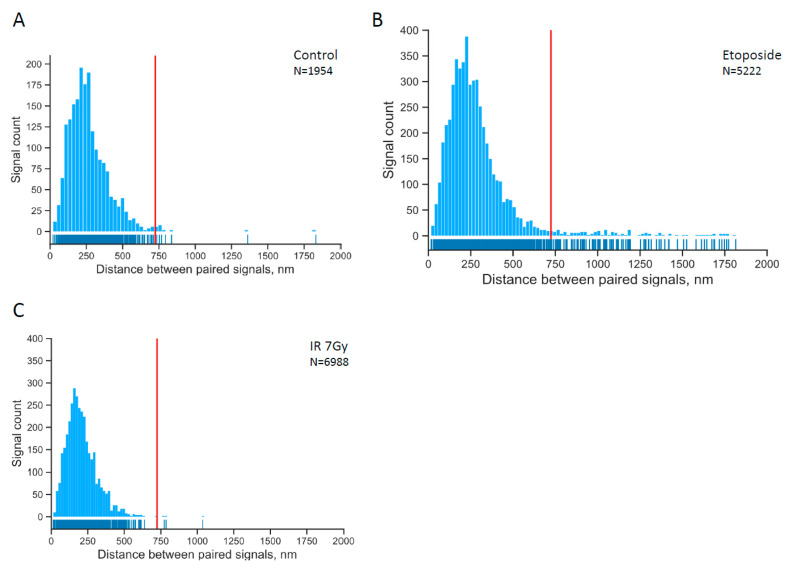
The distribution of pairwise distances within paired signals in control cells (**A**), in etoposide-treated cells (**B**) and cells treated with ionizing radiation (IR) (**C**). The red line separates the 99% quantile for the control experiment. The tail on the histogram of pairwise distances of signals in etoposide-treated cells is non-random with a confidence level of 0.997 (see Statistics in the Section 4).

**Figure 3 ijms-23-09824-f003:**
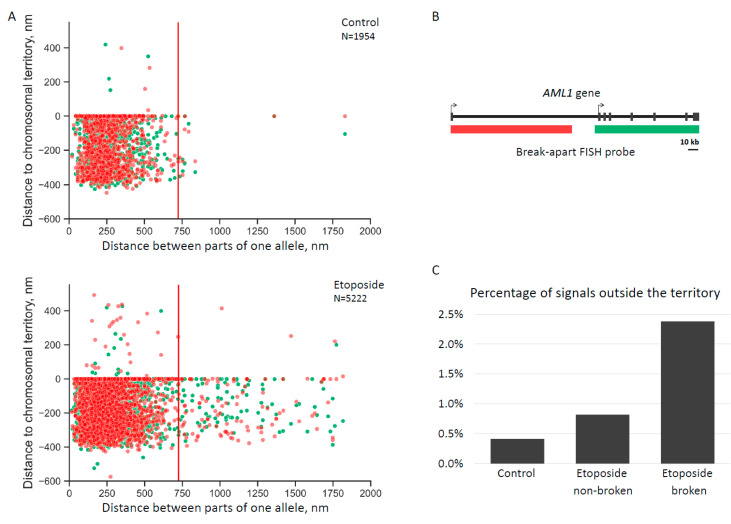
(**A**) Scatterplots reflect the distances between regions of one allele (x-axis) and the distance to the territory (y-axis, the value 0 reflects the border of the chromosome territory) for each *AML1* region. Red and green colors reflect upstream and downstream regions of the *AML1* break-apart FISH probe. (**B**) The upper scatter plot shows the data for the control cells, and the lower scatterplot depicts the etoposide-treated cells. N is the total number of dots in the scatterplot. (**C**) Percentage of alleles localized outside the chromosomal territory in control cells, as well as non-broken alleles and broken alleles in etoposide-treated cells (see statistic in Appendix A).

**Figure 4 ijms-23-09824-f004:**
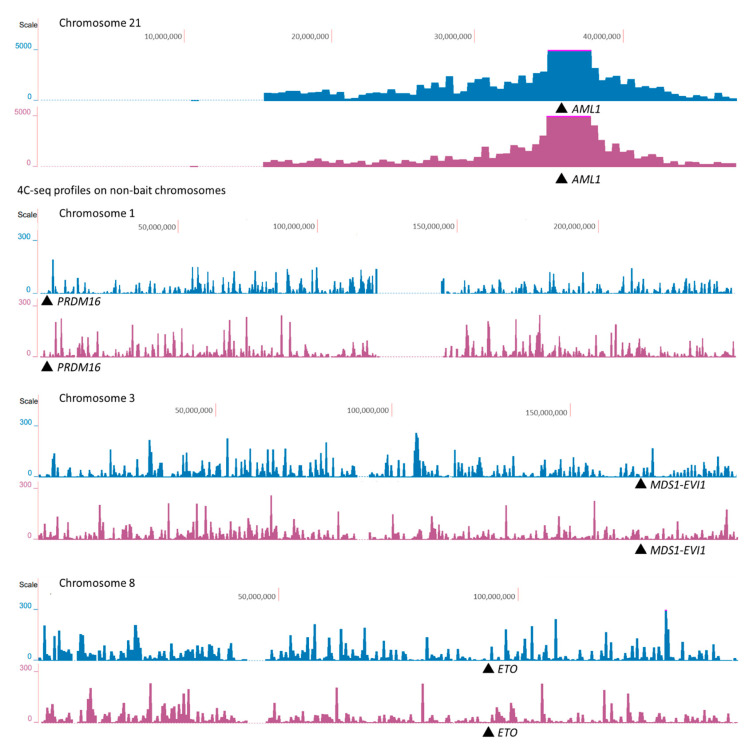
4C-seq profiles of *AML1* from untreated cells (blue) and from cells treated with etoposide (byzantium). The 4C-seq profiles of *AML1* on chromosome 21 show *cis*-contacts (first pair of rows), and the 4C-seq profiles of *AML1* on chromosomes 1, 3, and 8 show *AML1 trans*-contacts with chromosomes that bear *AML1* translocation partners in TI-related AML (other pairs of rows). *AML1* translocation partner genes are indicated by arrows. Data resolution is 500 kb. 4C-seq data are collected in Appendix A.

**Figure 5 ijms-23-09824-f005:**
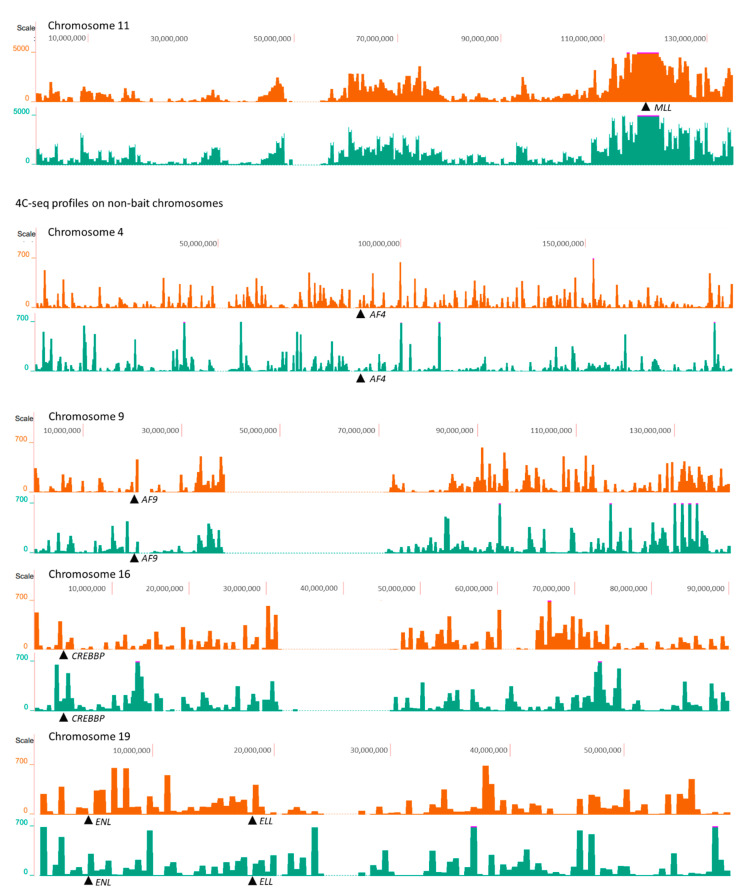
4C-seq profiles of *MLL* from untreated cells (orange) and from cells treated with etoposide (green). The 4C-seq profiles of *MLL* on chromosome 11 show *cis*-contacts (first pair of rows), and the 4C-seq profiles of *MLL* on chromosomes 4, 9, and 19 show *MLL trans*-contacts with chromosomes that bear *MLL* translocation partners in TI-related AML (other pairs of rows). *MLL* translocation partner genes are indicated by arrows. Data resolution is 500 kb. 4C-seq data are collected in Appendix A.

**Table 1 ijms-23-09824-t001:** Recurrent translocations in TI-related AML. The HUGO Gene Nomenclature Committee’s gene names are highlighted in bold.

Gene	Translocation	Partner	References
	t(8;21)(q22;q22)	*ETO (**RUNX1T1**)*	[13,14,19,20,21]
*AML1* (***RUNX1***)	t(3;21)(q26.2;q22)	*MDS1-EVI1 (**MECOM**, PRDM3)*	[22,23]
	t(1;21)(p36;q21)	** *PRDM16* **	[24,25]
	t(9;11)(p22;q23)	*AF9 (**MLLT3**)*	[14,20]
	t(4;11)(q21;q23)	*AF4 (**AFF1**, MLLT2)*	[26,27]
*MLL* (***KMT2A***)	t(19;11)(q13;q23)	** *ELL* **	[20,28]
	t(11;19)(q23;p13.3)	*ENL (**MLLT1**)*	[20,28,29]
	t(11;16)(q23;p13)	** *CREBBP* **	[30,31,32]

## Data Availability

3D-FISH data generated during the study can be downloaded at: disk.yandex.ru/d/WKXi5sWygeD6FA (accessed on 24 August 2022). The Conredet code is available at: https://github.com/dendik/conredet (accessed on 24 August 2022).

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
