# Peer review of "Recurrent Translocations in Topoisomerase Inhibitor-Related Leukemia Are Determined by the Features of DNA Breaks Rather Than by the Proximity of the Translocating Genes"

_ijms, 2022, doi:10.3390/ijms23179824_

Round 1

Reviewer 1 Report

Etoposide and other Top2-poisons induce AML1-MLL translocations thereby generating fusion-proteins that drive the development of t-AML. The cancerogenic process is known for decades, but the mechanism preferentially promoting the AML1-MLL -translocation is still unclear. The paper of Lomov et al addresses in this context the most crucial of unsolved questions, i.e. are these balanced translocations favored by spatial proximity of the involved loci?

Using 3D-interphase-FISH, chromosome territory painting and a break-apart probe for the AML1-break site they show that etoposide induces indeed a breakage of the AML1-locus and that a few percent of these broken ends fall apart and move outside of their normal chromatin domain and chromosome territory. A similar phenomenon has previously been demonstrated for the MLL-locus (ref. 43). The authors move on to demonstrate by 4C-seq that neither the unbroken nor the etoposide-exposed translocation loci have a significant predisposition to make contact with each other.

These data support the main conclusion of the paper that cancerogenic translocations following treatment with topoisomerase II poisons are not due to spatial proximity of the translocating genes but rather due to abnormal mobility and temporal persistence of the breaks.The paper thus provides an interesting set of experimental data that shed new light on that old story.

The paper is interesting, well written and merits publication. However, I have a few concerns with the experiments and presentation and interpretation of the data, which should be addressed.

Major concern 1

AML1-MLL translocations are a typical sequel of treatment with topo 2 poisons. They are rarely seen following other DNA-double strand break-inducing therapies (e.g. IR). Thus, the obvious question arises, how specific are the observed phenomena for etoposide and how do they compare to IR (creating short-lived DSB). Including an IR-control would greatly improve the paper.

Major concern 2

Work by Cowell et al (ref 7) has demonstrated colocalisation of TOP2-linked break-sites of AML1 and MLL-loci upon inclusion into the same transcription factory. Thus, one would expect that experiments such as presented here by Lomov et al possibly would yield different results in Mid-S-Phase than in late S or G2. Unfortunately, the authors have not at all addressed cell cycle distribution.

Minor concern 1

The uninitiated reader would greatly benefit from a more elaborate explanation of the 4C-seq technology, why it provides new insights into the matter that were hitherto not (or in a contradictory manner) obtained by other procedures. Most notably, the (negative) data presented in Fig. 4 and 5 should be more convincingly explained. The bald statements in lines 190-194 are rather unsatisfying. What is the 4C-seq positive control (i.e. how would a preferential contact look like, if it existed).

Minor concern 2

Line 293: “…the long persistence time of these “wandering” ends of DSBs that require a substrate for repair, even if the repair is erroneous.” Why do the authors discard the possibility of blunt-end repair?

Minor concern 3

The study is limited to Jurkat cells on plastique. Things may be different in real life.

Minor concern 4

Ref 6 an Ref 58 are identical.

Reviewer 2 Report

This is a very interesting manuscript that investigates an important caveat in the successful treatment of cancer, that of therapy-induced, in this case isomerase inhibitor induced secondary cancers. In general, the manuscript is well-written, the scientific quality of the introduction and discussion sections is excellent, results presentation and referencing are also of a high standard and figures are clear and informative. I have two major reservations regarding the manuscript overall, firstly the cell culture model used, and secondly the programme Conredet, for which there seems to be no prior, peer-reviewed publication. These questions and comments are summarised below.

1. The cell culture model used is the extensively cultured and malignant Jurkat cell line, which is exposed to 100ug/mL etoposide for just 1 hour. While the authors have published with this model previously, one would have to question whether this is an appropriate cell line or an appropriate etoposide treatment schedule to mimic the scenario that is observed in humans. In order to validate the model, is there a ‘positive control’ that the authors can report or reference where spatial proximity of rearrangement partners is able to be detected following 1 hour of etoposide treatment of Jurkat cells? Is it possible that the ‘negative’ result is due to the model and not because this mechanism does not apply?

2. Minor point. Have the authors karyotyped their Jurkat cells? The image in Figure 1 seems to indicate a high proportion of cells which are not diploid. What proportion of cells were able to be included in the analysis?

3. I cannot find reference to a prior publication describing Conredet. While development of new software and its availability on platforms such as github are important, the accuracy and integrity of the software needs to be validated and subjected to peer review. While the simplified descriptions provided in the methods section are appreciated by readers with expertise in biology (but not computing), this does not constitute a proper validation and peer review of the software. Could the authors please provide references to or evidence of review of the software and its validation?

4. Please check for self-plagiarism for parts of the methods section (I am not sure whether the level of similarity with the authors’ previous publication is within acceptable limits).

5. Please check for minor errors in citation of sources of reagents and minor typographical errors in references.

Round 2

Reviewer 1 Report

I am grateful to the authors for taking up all of my suggestions. The paper has been greatly improved. It is now well understandable, interesting and fit for publication.

Author Response

We sincerely appreciate all valuable reviewer comments, which helped us to improve the manuscript.

Reviewer 2 Report

The investigators have acknowledged reviewers’ comments, however not all of the comments have been addressed. Firstly, there is no indication that the cell culture model used by the authors produces the type of data (for any gene) that fulfils the investigators’ criteria for spatial proximity of rearrangement partners. This does not invalidate the results presented in the manuscript – in fact, verification of the model (cell line, etoposide treatment regimen) would further validate results.

Because I do not have the computing expertise to evaluate the Conredet programme, and rely on peer-reviewed publication to comprehensively describe and authenticate bioinformatics programmes, I am unable to judge the data produced by this programme. I feel that an alternate reviewer should be sought who can evaluate the algorithms and validation data for Conredet.

Author Response

We sincerely appreciate all valuable comments, which helped us to improve the  manuscript.

Same cell line and etoposide treatment regimen were used in our previous work [ref.65] and FISH showed a twofold change in the frequency of colocalization of the AML1 and ETO genes after etoposide treatment. This gave us a reason to perform 4C seq under these conditions. The 4C seq results showed no increase in the number of contacts of AML1 or MLL with translocation partners compared to other genes, which would be expected if partner genes did not just acquire random mobility in the nucleus but moved toward each other. We have expanded the discussion to make this clearer.